# A Review of the Neuropsychological Dimensions of Tourette Syndrome

**DOI:** 10.3390/brainsci7080106

**Published:** 2017-08-18

**Authors:** Simon Morand-Beaulieu, Julie B. Leclerc, Philippe Valois, Marc E. Lavoie, Kieron P. O’Connor, Bruno Gauthier

**Affiliations:** 1Centre de recherche de l’Institut universitaire en santé mentale de Montréal, 7331 rue Hochelaga, Montréal, QC H1N 3V2, Canada; simon.morand-beaulieu@umontreal.ca (S.M.-B.); valois.philippe@uqam.ca (P.V.); marc.lavoie@umontreal.ca (M.E.L.); kieron.oconnor@umontreal.ca (K.P.O.); 2Département de neurosciences, Université de Montréal, 2960 Chemin de la Tour, Montréal, QC H3T 1J4, Canada; 3Département de psychologie, Université du Québec à Montréal, 100, rue Sherbrooke Ouest, Montréal, QC H2X 3P2, Canada; 4Département de psychiatrie, Université de Montréal, 2900, boulevard Édouard-Montpetit, Montréal, QC H3T 1J4, Canada; 5Département de psychologie, Université de Montréal, Campus Laval, 1700 rue Jacques-Tétreault, Laval, QC H7N 0B6, Canada

**Keywords:** Tourette syndrome, tics, neuropsychology, attention, memory, motor skills, language, executive function, social cognition, academic performance

## Abstract

Neurocognitive functioning in Tourette syndrome (TS) has been the subject of intensive research in the past 30 years. A variety of impairments, presumably related to frontal and frontostriatal dysfunctions, have been observed. These impairments were found in various domains, such as attention, memory, executive functions, language, motor and visuomotor functions, among others. In line with contemporary research, other neurocognitive domains have recently been explored in TS, bringing evidence of altered social reasoning, for instance. Therefore, the aims of this review are to give an overview of the neuropsychological dimensions of TS, to report how neuropsychological functions evolve from childhood to adulthood, and to explain how various confounding factors can affect TS patients’ performance in neuropsychological tasks. Finally, an important contribution of this review is to show how recent research has confirmed or changed our beliefs about neuropsychological functioning in TS.

## 1. Introduction

### 1.1. Clinical Features of Tourette Syndrome

Tourette syndrome (TS) is a neurodevelopmental disorder characterized by both multiple motor and one or more phonic tics [1]. Onset of TS occurs during childhood, and tic frequency and severity is known to wax and wane [2]. Children and adolescents are the most affected by TS, with an estimated population prevalence of 0.3% to 0.9% [3]. Other tic disorders, such as persistent or transient tic disorder, have an higher prevalence, especially in childhood [4]. TS may also cause several other functional impairments that can affect quality of life; such as physical discomfort associated with tics, poor concentration, relationship problems (e.g., stigmatization, victimization), and academic, cognitive, and emotional problems that can result in explosive outbursts [5,6,7,8,9]. Investigators have reported mild to moderate functional social impairment and avoidance associated with tic severity [10]. In TS, comorbidity is a major feature and it has a role in the later psychosocial outcome and behavioral problems. Anxiety, oppositional behavior, and depressive symptomatology are commonly found in TS [11,12], but attention deficit hyperactivity disorder (ADHD) and obsessive–compulsive disorder (OCD) are the most common psychiatric comorbidities reported among TS patients and have multiple social and behavioral consequences that add up to those of TS itself [11,13]. Obsessive–compulsive symptoms (OCS) can also affect TS patients, even if they do not reach the required threshold for the diagnosis of OCD [14]. Subclinical OCD or OCS in TS patients often take the form of intrusive thoughts about symmetry, ordering, aggression or sexuality, but more rarely involves fear of contamination, which is typical of OCD [15]. OCS also occur more frequently in TS patients than clinical OCD reaching diagnosis threshold [11,16,17].

Over the past 30 years, much research has been conducted on the neuropsychological functioning of patients affected by TS. Neuropsychology is a discipline that examines various types of complex relationships between the functioning of brain structures and higher mental neurocognitive operations. Recent progress in neuroimaging, electrophysiology, and genetics have allowed deciphering an important portion of the neurobiological mechanisms underlying TS [18,19,20].

### 1.2. Neurobiological Origins of TS

Much of the cognitive, behavioral, and emotional impairments affecting TS patients are linked to cortico–striato–thalamo–cortical (CSTC) circuits [21,22,23,24]. These circuits assure the communication between subcortical structures, such as the basal ganglia, and the cortex. The basal ganglia receive inputs from the whole cortex. These inputs are mostly motor, limbic and associative. They stay parallel throughout their projections to the basal ganglia, and this segregation leads to the advent of functional loops: the CSTC circuits. However, there are some cross-connections between different circuits, which allow the integration of information. Behavior regulation as a function of a given context is granted by this integration [25,26]. Five different CSTC loops have been identified: motor, oculomotor, prefrontal, orbitofrontal, and cingulate circuits [27]. These circuits are involved in multiple cognitive functions such as planning, movement execution and inhibition, motivational regulation of behavior, error detection, and associative learning. Each of these circuits is divided in two loops: the cortico–striatal and the cortico–subthalamic circuits [28]. The first receives input from the whole cortex, while the latter receives almost only motor input from the frontal cortex. Impaired functioning of the cortico–striatal circuit might lead to reduced inhibition from thalamus nuclei, which could result in involuntary movements. In this vein, Mink [21] proposed that tics might be caused by unexpected activation of striatum cells, which would lead to the production of unwanted movements. Furthermore, thinning of the sensorimotor cortex was found in TS patients and correlated with tic severity [29]. A smaller volume of caudate nuclei was also reported, which is consistent with the hypothesis of impairments in CSTC circuits [22]. The neurological anomalies mentioned above seem to cause impairments beyond the motor and phonic tics. Indeed, they have consequences on the neuropsychological functioning of TS patients. Research on this matter has produced numerous but sometimes conflicting findings. 

### 1.3. Toward a Distinctive Neuropsychological Profile of TS

In complement with studies investigating brain functions at the origin of tics, neuropsychological research on TS yielded important findings, but discrepancies still exist throughout the scientific literature, caused by various confounding factors. These confounding factors can be understood as the important heterogeneity in patients’ symptoms and associated features, which are not always controlled in some studies. The principal challenge remains in the disentanglement of multiple comorbid disorders superimposing over symptoms of tics. For example, ADHD and OCD are each reported in at least 20% of TS patients’ cases [3,30], but could be even more frequent in clinical samples [11]. These features could be associated with increased neurocognitive impairments, behavioral problems, and learning disorders [30]. Therefore, an important reason for the discrepancies in neuropsychological studies in TS could be the exclusion or not of comorbid disorders as well as how these disorders were assessed and diagnosed. Another factor that can considerably influence the results of neuropsychological studies is the frequent inclusion of patients under psychiatric medication [31,32]. Other treatments, such as cognitive–behavioral therapy (CBT) or deep-brain stimulation (DBS), can affect neuropsychological functions as well [33,34]. Furthermore, given its impact on brain maturation and symptoms’ expression, patients’ age could also explain some of the differences between studies. Finally, as TS is more common in males than in females, and as males are more likely to have higher rates of comorbidities, gender could also be a confounding factor in modulating the neurocognitive functioning [35].

A review on the neuropsychological aspects of TS was conducted by Eddy et al. [36] in 2009. The authors identified study limitations as stated previously: problems with measures and confounding factors (the influence of age, tic severity, and comorbidities on cognitive performance). However, they provisionally concluded that there is considerable evidence for cognitive impairments that seem to be intrinsic to TS, and that these impairments may reflect dysfunction of the anterior cingulate network within the frontostriatal pathway [36]. They specifically found robust evidence for inhibitory deficits on certain tasks, considerable evidence for deficits in motor skills and visuomotor integration, and equivocal evidence (because of the presence of comorbidities in the studied samples) for deficits in fluency, planning, working memory, cognitive flexibility, attention, and memory. Since this review was conducted almost 10 years ago, we now need to know if these results were confirmed and if the many new studies, with an improved study design, shed a different light on this distinct and complex problem. With the major theoretical breakthroughs of the past few decades, neuropsychology still represents a discipline strategically well positioned to understand both brain functioning and functional impairment encountered in TS. By merging recent advances with prior knowledge, this review will yield a comprehensive picture of the neuropsychological functions of TS patients. Thus, the current review aims at (1) updating the state of knowledge in the field of neuropsychology in TS, (2) reporting how neurocognitive functions evolve during the lifespan of TS patients, (3) and presenting other factors that might modulate impairments in these functions.

## 2. A Review of Neuropsychological Functioning in TS

### 2.1. Intellectual Abilities

Despite the long-standing belief that TS patients have normal intelligence as measured by intellectual abilities tests [37], a recent cohort study conducted in Denmark reported a lower global IQ of almost 12 points in TS children, as measured with the Wechsler Intelligences Scale for Children—3rd edition (WISC–III [38]). This was characterized by both lower verbal comprehension and performance IQs [39]. In this study, lower IQ was associated with earlier onset of tics and the presence of comorbid disorders. Khalifa et al. [40] also reported slightly below average verbal and performance IQ in a population-based study of school-age children, also noting important variations among patients full-scale IQ, ranging from mental retardation to two standard deviations above mean. Yet, mental retardation has only been reported in 4% of TS patients [11,13]. While there might be significant variance in intelligence across TS patients, only slight impairments in general intelligence are usually reported. One possible explanation for these differences could reside in patients’ performance during intelligence tests, which could be impaired by the presence of tics [41] or by various comorbid disorders. However, mixed results have been reported regarding the impact of comorbid disorders on IQ. On the one hand, the presence of comorbid ADHD has been associated with a lower IQ [42,43], while comorbid OCD has been associated with a higher IQ [43,44] in TS patients. On the other hand, others have found that in TS patients, only the combination of ADHD and OCD worsened IQ scores; and that patients with either comorbid ADHD or OCD would not differ from TS-only patients on global IQ [39]. A similar result was reported by Khalifa et al. [40], who did not find an impact of ADHD on the global IQ of TS patients. These conflicting results may stem from the use of different IQ assessment methods. Age also seems to be a confounding factor, since earlier onset of tics is associated with lower IQ [39]. To our knowledge, all research on this matter was conducted in children and adolescents, and little is known about intellectual abilities in adults with TS. Therefore, future research should aim at understanding how patients’ age can impact their IQ.

### 2.2. Attention

Attention is a multidimensional phenomenon that cannot be conceptualized as a single entity. It is based on the existence of several processes of selection. These subsystems perform different functions, which are interconnected in the concepts of orienting, shifting, selective, divided, and sustained attention (or vigilance) [45,46]. Given the frequent comorbidity observed between TS and ADHD [3,11], attentional deficits are often encountered in TS patients.

#### 2.2.1. Orienting and Shifting of Attention

Orienting attention involves the ability to direct attention at a given location and to reorient or shift attention to a new location [47]. In a dichotic listening task, TS children and adolescents did not differ from healthy controls regarding the ability to shift attention to focus on right ear stimulus. However, their performance decreased when asked to focus on left ear stimuli, suggesting altered corpus callosum functioning in TS [48]. Yet, only 7–19 years old boys participated in this study and the sample of TS patients was quite small (*n* = 20), which limits its generalizability. 

Within the tactile modality, adults with TS do not have problems in orienting attention towards a target location [49,50], despite being slower (between 1 and 1.5 SD) than healthy controls when shifting the focus of attention [51].

#### 2.2.2. Selective Attention

Selective attention can be defined as the capacity to focus on significant information and to ignore insignificant information [47]. While orienting of attention and selective attention have sometimes been used interchangeably, there might be some necessity to differentiate them, since they activate different brain regions [52,53]. Also, orienting of attention requires only target detection, while selective attention requires target identification [47]. The D2 cancellation task mainly assesses sustained attention, but it also assesses selective attention through participants’ ability to focus on and select target stimuli [54]. To our knowledge, only one study used this task to assess attention in children with TS, and reported normal performance [55].

In a small sample of adults with TS + OCD (three of whom also had ADHD), Muller et al. [56] found an increased number of false positives in these patients during a D2 cancellation task. However, this performance was counterbalanced by a reduced number of omission errors during the task, suggesting a slower but more accurate performance. In another letter cancellation task, which involved single and double-letter variants, adults with TS tended to show delayed responses in the more cognitively demanding task (i.e., the double-letter variant) [57]. Using an attentional blink paradigm, Georgiou–Karistianis et al. [58] showed that there was a preservation of selective attention functioning in adults with TS. However, those patients tended to make more post-target intrusion errors, suggesting a possible cognitive processing irregularity (i.e., different strategies to retrieve information or to integrate it) in TS, rather than an attentional or motor impairment *per se*. These results suggest that subtle cognitive processes beyond selective attention, such as reduced error monitoring or processing speed, are possibly impaired in TS. Yet, patients’ age and comorbidities need to be considered to get a clearer picture of selective attention abilities in TS.

#### 2.2.3. Divided Attention

Divided attention is often understood as the ability to optimize resource allocation in relation to environmental requirements [59]. Tasks used to measure this ability usually involve stimuli from two different dimensions or modalities [60]. In the auditory consonant-trigram task, children with TS tend to have poorer divided attention than children with OCD or healthy controls [61]. In other studies using dual-task paradigms, impairments were usually found in TS + ADHD children [62], or TS children with various comorbidities (mostly ADHD and OCD) [63], but not in TS-only children. 

In adults with TS, Johannes et al. [64] found deficits in dual-task performance that were specific to auditory stimuli, when highly conflicting visual stimuli were presented concurrently. However, patients with comorbid disorders such as ADHD or OCD were not excluded in their study. Nevertheless, a small sample of adults with TS + OCD showed no divided attention impairments [56]. While comorbid OCD does not seem to be associated with divided attentions deficits, the presence of ADHD could be at the source of such impairment, rather than TS itself.

#### 2.2.4. Sustained Attention

Sustained attention can be conceptualized as the capacity to maintain attention over time to detect infrequent signals [47], and is often measured with a continuous performance test (CPT). Therefore, omissions in CPT can yield precious information regarding sustained attention in TS patients, and have been associated with poorer tic suppression abilities [65].

Sustained attention deficits have been commonly reported in children with TS, mainly as omission errors during a CPT [55,62,66,67,68,69,70], but also as longer CPT reaction times (RT) [62,71]. Here, comorbidity could impair sustained attention abilities, as normal performance in TS-only children was often reported [68,69,72,73]. Specifically, ADHD could be an important confounding factor, since TS + ADHD children made more omission errors than TS-only children [32,68,70,74] or healthy controls [68,69,70]. However, other studies went against this trend and did not find impaired sustained attention in TS + ADHD children [72,73,75,76]. OCD could also impair TS patients sustained attention, since more omissions were reported in TS + OCD than TS-only children [77]. Furthermore, the authors of this study reported that an additional diagnosis of ADHD in those patients worsened their attentional performance. Using a 2 × 2 factorial model, Uebel–von Sandersleben et al. [78] found that sustained attention was impaired by the ADHD factor, but not the TS factor. However, impairments in sustained attention were also found in TS-only children [70], suggesting that some attentional deficits could be inherent to TS. Yet, the available data confirms the role of ADHD in potentiating attentional deficits in children with TS. 

In adults with TS, few studies assessed sustained attention. Normal CPT performance was found [79] and, as reported before, studies using a D2 cancellation task did not find an increased omission rate [56]. However, impairments were found in a highly demanding letter cancellation task [57]. Furthermore, Georgiou et al. [50] reported impaired sustained attention during a vibrotactile task. The small amount of studies assessing sustained inhibition in adults with TS prevents us from making a strong assumption regarding this function. It is possible that some impairments are still present in adults, but more research is needed.

#### 2.2.5. Attentional Capacities of TS Patients

All in all, the available evidence suggests that TS patients present slight impairments in attentional capacities. These impairments could, however, be caused by tics or by efforts to inhibit them [41]. It appears that ADHD acts as a major confounder regarding the attentional capacities of TS patients. Hence, intact attentional performance was often reported in patients without ADHD, either for selective [55], divided [62,63] and sustained attention [68,69,72,73]. Therefore, ADHD in TS patients would come with marked deficits in attentional capacities. Yet, the impact of comorbid OCD on attentional capacities is not fully understood. Normal CPT performance was reported in OCD patients [80,81], but Lucke et al. [77] suggested that the deficits in sustained attention could represent a core marker of TS + OCD, compared to either TS or OCD alone. Intrusive thoughts could possibly alter attention in TS + OCD patients, but more data are needed to corroborate this hypothesis.

Task complexity could explain some of the discrepancies between studies. For example, normal attentional performance was reported in a traditional D2 cancellation task [55] but impairments were found in a more demanding version of this task [57]. Studies that did not exclude psychiatric medication could also mix up things, since guanfacine has been proved to lower omission errors in TS + ADHD children [31].

### 2.3. Memory

Memory allows us to encode, store, consolidate and retrieve a certain quantity of information. The contemporary model of memory divides it into short-term working memory and long-term memory [82,83,84,85]. Working memory stores information that is kept for only a few seconds to a few minutes. The long-term memory processes can store information on much longer time periods, ranging from a few days to several years. Many reports suggest possible impairments in working as well as long-term memory in TS.

#### 2.3.1. Working Memory

In children and adolescents with TS, good verbal working memory was often reported, as proven by normal forward and backward digit span [61,86,87] and good *n*-back performance [88]. The presence of ADHD could impair working memory capacities, since worse forward and backward digit span was found in TS + ADHD children, when compared to TS children without comorbidities [89]. Nevertheless, impaired verbal working memory performance was also found in TS-only children, who had deficits in forward digit span [63], suggesting that verbal working memory deficits could exist in some non-comorbid TS patients. However, the sample of TS-only patients in this study was rather small (*n* = 11), making it hard to draw conclusions on this base. Finally, letter number sequencing is not affected in children with TS [90], which can be explained by the fact that much of the variance in this test is explained by digit span [91].

Visual and visuospatial working memory could be impaired in children with TS, as suggested by impaired spatial span and worse performance at the 8-box task [67,71]. However, normal performance has been reported on the Corsi span [92], the Benton visual retention test [92], the delayed matching to sample task [67], the self-ordered pointing task [92], and the finger windows (a subtest of the wide range assessment of memory and learning) [61]. Comorbid ADHD could lead to some deficits here again, since Termine et al. [87] found impaired forward and backward Corsi span in TS + ADHD but not in TS-only children.

In adults, verbal working memory generally seems to be unimpaired, as indicated by normal forward and backward digit span [93,94], and good *n*-back [95] and digit ordering [96,97] performance. However, adults with TS + OCD made more omissions in a 2-back task than healthy controls [56]. Here, comorbidity could explain some of the differences between studies, since Channon et al. [95] reported no impairment in the *n*-back task for non-comorbid TS adults, either with the 1-back or the 2-back variant. Some studies found impaired working memory in the digit ordering task [98,99]. One of these studies included only non-comorbid TS adults [99], therefore comorbidity cannot account for all verbal working memory impairments reported. As in children with TS, letter number sequencing is not affected in adults [91].

Regarding visual and visuospatial working memory in adults with TS, Channon et al. [57] found an intact backward Corsi span but impaired forward Corsi span. Intact performance was, however, reported in the Benton visual retention test and the self-ordered pointing task [93]. 

While working memory generally seems to be intact in children and adults with TS, some studies reported conflicting findings. TS symptoms and severity of comorbid disorders could explain some of the discrepancies between the results, since those factors are good predictors of visuospatial working memory performance [89]. Tic severity could also explain some of the reported discrepancies, since TS children with moderate tic symptoms (Yale Global Tic Severity Scale (YGTSS) = 48) had worse oculomotor *n*-back performance than TS children with low tic severity (YGTSS = 24) and healthy controls [100].

#### 2.3.2. Long-Term Verbal and Nonverbal Memory

Verbal memory in TS appears to be intact. Indeed, children with TS had average performance during the word list of the California verbal learning test for children (CVLT–C [61,101]) and a test of story recall [102]. However, despite similar CVLT-C scores in TS children and healthy controls, Mahone et al. [103] found more CVLT-C intrusion errors in the TS group. This result seems logical for TS patients, since intrusion errors represent poor inhibitory control (see Section 2.6.1). A trend toward an impaired performance at the word list Rey auditory verbal learning test (RAVLT [104,105,106]) was also reported [102] in children with TS. Similarly, normal CVLT performance was found in TS adults [107]. However, impairments were reported in another word list test [94] as well as in story recall [108].

Nonverbal memory deficits have been reported in TS. Indeed, poor performance on the Rey–Osterrieth complex figure (ROCF [104,109]) immediate and/or delayed recall have been found in both TS children [110,111,112] and adults [107,113]. These findings could arguably be related to other collateral processes such as the visuospatial and visuomotor impairments sometimes found in patients with tics, which could interfere with recall performances at the ROCF. These nonverbal memory impairments could also imply right hemisphere dysfunction, since patients with such alteration are usually more affected in delayed recall of the ROCF [114]. This would be consistent with anomalies in brain lateralization due to striatal damage often observed in TS patients [115,116]. Furthermore, Channon et al. [102] reported a trend toward impaired performance in the Visual Reproduction subtest of the Wechsler memory scale in children with TS. However, discrepancies still exist, as Chang et al. [61] reported normal ROCF recall in TS children.

These results suggest that a nonverbal memory deficit could be present in TS, which might be related to more general right hemisphere dysfunction, although this remains to be more thoroughly investigated. Regarding verbal memory, there is too much discrepancy, possibly due to different tests evaluating different memory constructs, and not enough data to draw a conclusion. 

#### 2.3.3. Recognition Memory

Few studies assessed the recognition memory of TS patients. Children with TS had worse performance than healthy controls in the spatial recognition memory subtest of the Cambridge neuropsychological test automated battery (CANTAB) [67]. TS children also had slightly reduced performance in the pattern recognition memory subtest, but that result did not reach significance level [67]. Using the same tests, Watkins et al. [117] reported impaired spatial and pattern recognition memory. Regarding verbal memory, word recognition was shown to be preserved in adults with TS [94]. Combined with the previous findings on long-term memory, these results suggest an impaired memory processing for nonverbal material, while verbal memory seems be intact.

#### 2.3.4. Implicit Memory

To our knowledge, only few studies assessed implicit memory in TS. Normal performance in stem completion, mirror reading and serial reaction time task was reported in children with TS [102]. Another study, using a process dissociation procedure, found normal implicit memory in TS children [118].

#### 2.3.5. Memory in TS Patients

Globally, TS patients do not appear to have widespread impairments in memory. Few studies pointed toward possible deficits in nonverbal memory, but this remains to be confirmed. Other domains, such as working and implicit memory appear to be intact in TS patients. Yet, factors such as comorbidity or tic severity could cause memory deficits in some patients.

### 2.4. Motor Functions and Spatial Cognition

Motor impairment seems relatively obvious in TS, since motor and phonic tics involve muscle contractions [119,120]. However, all motor functions are not equally impacted in TS. Recent findings pinpoint a chronic over-activation in cerebral regions associated before and during tic generation [121] and also during motor execution and inhibition in TS patients [122,123] which is likely to interfere with motor skills and execution of voluntary movements.

Various neuropsychological tools, such as the Purdue Pegboard [124], Grooved Pegboard, Finger tapping, and hand steadiness tests, as well as other motor processing tests were used to assess motor functions in TS patients.

#### 2.4.1. Motor Skills

Mixed findings have been reported regarding TS patients’ motor skills. In children with TS, some studies reported normal performance at the Purdue Pegboard [69,125], which measures fine and gross motor dexterity and coordination [124]. However, Bloch et al. [112] reported that children with TS performed one-half to one standard deviation below normative data, and that a deficit with the dominant hand predicted worse tic severity in adulthood. Worse performance was also found in children with more severe tics, when compared to less severe patients [126]. In the Grooved Pegboard test, which also assesses fine motor dexterity, impaired [127] or a trend toward impaired [128] performance was found in children with TS. To our knowledge, only one study used the pursuit rotor and mirror tracing tasks in TS children, and found normal performance in both tasks [129]. Few studies assessed fine motor speed with the Finger tapping test, and both intact [125,127] and impaired performance were found [68]. In simple reaction time tasks or tests measuring response speed, TS children are usually not slower than healthy controls [67,71]. Furthermore, children with TS had shorter movement time but not reaction time, which suggests an urge to respond in TS patients [71]. This is consistent with the reported accrued impulsivity among TS patients [130,131]. 

In adults with TS, most studies found Purdue Pegboard impairments [33,72,132,133,134], but normal performance was also reported [107]. Importantly, the study of Abramovitch et al. [134] had one of the largest sample among the neuropsychological literature in TS, and reported that patients performed one standard deviation below the norms on the bimanual Purdue Pegboard. Bornstein [135] found impaired Grooved Pegboard performance for the dominant hand, but not for the non-dominant hand. However, a more recent study reported normal Grooved Pegboard test performance in adults with TS [33]. Conflicting findings were also reported for hand steadiness, since a study found impaired hand steadiness [136], while another found normal performance in the Hole type steadiness test [33]. In the same vein, a recent study reported normal pursuit rotor task performance and the mirror tracing task in TS [129], while an earlier study found impaired performance on the same task [94]. Most studies in adults with TS reported intact Finger tapping performance [132,135,136], even when patients had comorbid OCD [56]. Normal performance was also reported in a simple reaction time task [132].

Medication and/or comorbidity could partially explain these discrepancies, since normal performance was found in non-medicated and non-comorbid TS patients [107,125]. However, impairments were also found in non-comorbid patients [133]. Few studies found no impact of comorbidity on the Purdue Pegboard [90], the Grooved Pegboard [90,137] and the Finger tapping test [137]. While performance on the Purdue Pegboard was shown to improve following CBT [133], a recent study showed that performance on this task was unrelated to symptoms improvement following CBT in a very large sample of adults with TS [134]. This difference could be explained by the use of different therapies. While Abramovitch et al. [134] used the comprehensive behavioral intervention for tics (CBIT) and psychoeducation and supportive therapy (PST), Lavoie et al. [133] used the cognitive–psychophysiological (CoPs) therapy, which aims to reduce overall motor activation and muscle tension.

Therefore, as reported by Kalsi et al. [138], it is hard to draw clear conclusions regarding fine motor skills in TS patients. Medication use, comorbidities and age of patients are confounding factors, but discrepancies remain even when taking these factors into account. The variety of tasks used to assess fine and gross motor skills in TS could also be responsible for some of the conflicting findings across different studies. We also encourage future studies to use larger samples, in order to provide robust results. 

#### 2.4.2. Visual Motor Integration and Visuoconstructive Abilities

Visual motor integration (VMI), which can be conceptualized as the ability to combine visual and motor information, is often been assessed with the Beery-VMI test [139]. Despite early findings revealing deficits in Beery-VMI for children with TS [72,128], more recent research suggests that TS patients do not exhibit consistent impairment at this test [61,66,69,89]. In a rather larger cohort, Baglioni et al. [140] found VMI impairments in 16% of TS children, suggesting that visuomotor deficits could exist in a small subset of the TS population. Visuoconstructive skills in TS children have also been assessed with the copy stage of the ROCF. Here, significant discrepancies exist, as both normal [61,110] and impaired [72,111,112] performance have been reported. Comorbid ADHD could be a exacerbating factor, since TS-only children have been shown to outperform TS + ADHD children on this task [74,141].

Few studies have assessed visuoconstructive functions in adults with TS, but Lavoie et al. [107] reported normal performance on the copy of the ROCF. However, Gruner [113] showed that adults with TS + OCD showed greater impairment in copying the ROCF than OCD-only adults. Interestingly, Randolph et al. [126] reported that tic severity does not impact the copy stage of the ROCF, which suggests that motor dysfunctions (i.e., tics) are not related to visuoconstuction. Here, comorbid disorders, rather than tic severity, seem to impair visuoconstruction.

### 2.5. Language

#### 2.5.1. Expressive Language and Speech

Verbal fluency, which is conceptualized as the ability to retrieve verbal content from episodic verbal memory, has mostly been assessed in TS with the controlled oral word association test (COWAT [142]). The test requires to produce, in a limited amount of time, as many words as possible starting with a given letter (letter fluency) or belonging to a given category (categorical fluency). The number of words in each condition is predicted by updating abilities and the number of words in the category subtest is specifically associated with lexical access speed [143], indicating that verbal fluency loads not only on vocabulary, but on executive function as well.

In TS, letter fluency seems to be preserved, as illustrated by intact performance in both children [73,74,86,89,103,144] and adults [34,56,93,94,117,145]. However, some studies found impaired letter fluency in children [40] and adults [99]. Few studies also revealed slight or trend toward impairments in letter fluency in children with TS [102,146]. Comorbidity could be a confounding factor, since TS-only children had better performance than those with TS + OCS and TS + ADHD + OCS [137]. However, normal performance was also reported in TS + OCD patients [56]. Moreover, two studies reported intriguing findings, with TS-only children outperformed by TS + ADHD and healthy controls [35,141]. However, impaired performance was found in 40 TS-only adults [99], which makes the situation even more puzzling. It is possible that letter fluency impairments only exist in few TS patients, as 17% of the sample of Zapparoli et al. [147] had impaired letter fluency. More studies with sound methodology and good control of confounding factors are needed to better pinpoint this ability in TS patients.

There are fewer discrepancies in results regarding categorical fluency, since most studies report intact performance in children [89,103,141] and adults [34,94,117,145,147] with TS. To our knowledge, only one study reported impaired categorical fluency [40]. In this study, the group of TS children scored in the lowest 25th percentile on categorical fluency. This result may have been confounded by comorbid disorders and/or psychiatric medication intake, since 68% of the group had comorbid ADHD and 44% were taking psychiatric medication. This is consistent with other studies that found worse categorical fluency performance in TS + ADHD children [74,148]. Although an early study found that tic severity was not related to categorical fluency [126], more recent results suggest that TS symptoms are a good predictor of verbal fluency [89]. 

Phonological and morphological processing also seems to be intact in TS patients. Compared to healthy controls, children with TS had normal accuracy and were faster in a non-word repetition task [149], and were also faster at producing the past-tenses of regular verbs in a past-tense production task and at naming manipulated items in a picture naming task [150].

Stuttering is not frequent in TS, as it affects less than 10% of patients [11]. However, slight impairments in speech fluency might exist. Indeed, typical disfluencies, such as hesitations, word repetitions and pauses, occur more frequently in TS patients [151]. Future studies should assess the link between such disfluencies and phonic tics in TS patients.

#### 2.5.2. Language Comprehension

While much research has been conducted regarding TS patients’ speech and language production, there are only a few studies that assessed receptive abilities. TS patients do not seem to have general difficulties with formal language comprehension, but pragmatic language impairments have been found in regards to understanding nonliteral remarks, such as metaphors and sarcasm [152]. Furthermore, Drury et al. [153] showed that impairments were limited to indirect sarcasm, where the remark differs from its literal meaning but is not directly opposed to it, while remarks in direct sarcasm are directly opposed to their literal meaning. Indirect sarcasm is therefore more subtle and complex. Authors mentioned these impairments are unlikely to be influenced by TS symptomatology, comorbid disorders, or medication intake. Such impairments in indirect sarcasm comprehension could also emerge from deficits in social cognition (see Section 2.7).

### 2.6. Executive Functions

Executive functions are a set of processes that facilitate the adaptation to new or unusual situations, for which no preexisting solution already exist. They are an aggregate of cognitive processes and behavioral abilities, including reasoning, planning, problem solving, cognitive flexibility and response inhibition, among others [154]. These functions involve prefrontal and subcortical structures, which interact with all associative areas in the brain [155]. Abnormal connectivity between motor cortical areas and basal ganglia related to cognitive control and premonitory urges has been identified in TS [18], which could impair patients’ executive functions. 

#### 2.6.1. Inhibition

The elaboration of inhibitory control, which can be defined as the ability to suppress or neutralize distractors that can interfere with attentional resource allocation, is an important aspect of cognitive development [156]. Research in response inhibition processes in healthy controls pinpointed the role of dorsolateral and orbitofrontal cortices in successful inhibition of meaningless stimuli, responses and impulses [157,158,159,160].

As reported by prior reviews of neuropsychological functioning in TS [36,161] and a recent meta-analysis on inhibitory control [130], verbal inhibition is impaired in TS patients. Deficits in the Hayling sentence completion task have been commonly found in TS patients [96,102,145], even in those without comorbidity [88,95,108,162]. Those deficits in non-comorbid patients could indicate impaired inhibitory control that is independent of ADHD symptoms [36]. However, a recent study found normal Hayling performance in adults with TS [98]. Yet, this study had a small sample size (*n* = 20), and 75% of the sample was under medication. Such limitations could explain this discrepancy with most evidence that pointing toward impaired verbal inhibition. 

The picture is less clear regarding other inhibition tasks, such as the Stroop task. On the one hand, several studies reported intact Stroop performance in children [69,86,91,144,148,163,164,165,166] and adults [57,79,91,107,123,145,167] with TS. On the other hand, impaired Stroop performance has also been found in TS children [61,168] and adults [34,56,93,97,99]. In a few occurrences, these impairments only concerned comorbid patients [102,169], but impairments have also been found in TS-only patients [97,99]. In a study where deficits in the Stroop task were found, results stayed significant even when covarying with ADHD symptoms [61]. A meta-analysis of 25 TS studies revealed moderate but significant impairments in Stroop performance [130]. Since the deficit is not severe, studies reporting a normal Stroop performance might have lacked the power to detect a difference between TS patients and healthy controls.

In the same vein, stimulus-response compatibility paradigms, such as flanker or Simon tasks, can also be used to study inhibitory functions [170,171]. As in the Stroop task, these paradigms induce conflicting responses, where participants must inhibit the inappropriate and automatic response to produce the required response [171]. However, it is not clear how TS patients behave in these tasks, as both normal [172] and impaired [88] flanker performance have been reported in children with TS. In adults, most studies reported normal flanker performance [95,173]. In Simon tasks, normal performance is often reported in both children [174,175,176] and adults [122,123]. However, impaired Simon task performance has also been found in children [177] and adults [178] with TS. In their study, Tharp et al. [177] reported that tic severity was negatively correlated with accuracy and positively with response time (RT). Such correlation between RT and tic severity was also reported in the study of Baym et al. [175], suggesting that while TS patients’ performance in this type of task is usually intact [130], patients with severe tics could experience some difficulty.

In TS, motor inhibition has often been studied with go/no-go tasks. However, only few studies suggested go/no-go impairments in TS patients. Goudriaan et al. [179] found more commission in adults with TS, and Greimel et al. [75] reported a trend toward more commissions in children and adolescents with TS + ADHD. However, while testing medication-free and non-comorbid TS patients, many studies reported no difference between TS patients and healthy controls during this type of task [62,164,176,180,181]. Even in studies including comorbid patients, go/no-go performance in TS patients did not significantly differ from healthy controls [56,117,182,183,184]. The intact go/no-go performance has been confirmed by a meta-analysis of 14 studies [130]. Also, some studies reported normal go/no-go inhibition with slower RT, suggesting a compensatory slowing of motor output to facilitate tic control [185,186,187].

Commission errors in a CPT can also be useful to assess motor inhibition in TS [188]. Few studies reported impairments in TS children [67,71,72,189,190] and adults [34]. Furthermore, CPT commissions seem to be frequent in TS + ADHD patients [32,70,74,141]. While normal CPT performance has also been sparingly reported in children [73] and adults [79] with TS, it has been confirmed that both groups exhibit moderate, yet constant deficits [130]. Medication could be a confounding factor here, since both guanfacine [31] and pimozide [32] have been shown to decrease CPT commissions in TS patients.

Oculomotor studies may also yield pertinent information about response inhibition. On the one hand, some studies reported increased latencies [100,191,192,193] or higher error rate [191,193,194] for TS patients during antisaccade tasks. Surprisingly, tic severity is unlikely to affect oculomotor response inhibition, since Jeter et al. [100] reported a trend toward longer latencies for TS children with low severity when compared to with children with moderate severity. On the other hand, enhanced cognitive control in antisaccade tasks has also been reported. In the study of Tajik-Parvinchi and Sandor [195], the TS + ADHD + OCD patients had shorter antisaccade latencies than healthy controls, TS-only and TS + ADHD patients, and made less errors than healthy controls and TS + ADHD patients. Some studies also reported that children with TS could switch between prosaccade and antisaccade tasks faster than healthy controls [196,197]. Another study found similar switching latencies for TS children and healthy controls, but better switching accuracy for TS children [198]. Therefore, more research is needed to better understand how TS patients perform during oculomotor paradigms. 

The presented data suggests selective impairments in response inhibition. Indeed, deficits were found in verbal inhibition, Stroop and CPT performance, but intact go/no-go and stimulus-response compatibility were also reported. Since some components of response inhibition are affected in TS, it could impair performance in other tasks. For example, Muller et al. [56] reported that TS + OCD patients made more errors in attentional tasks, which could be attributable to poor monitoring and inhibitory capacities, rather than to attentional deficits *per se*.

#### 2.6.2. Cognitive Flexibility

Cognitive flexibility can be defined as the capacity to adapt cognitive strategies to face new or unexpected situations [199]. It is thought to involve the orbitofrontal cortex, which also plays a role in inhibitory control [200,201]. The Wisconsin card sorting test (WCST [202]) is one the most used neuropsychological test to assess cognitive flexibility. In TS, most studies report normal WCST performance, both in children [89,141,166,169] and adults [33,56,93,107,108,135,162,203]. Yet, few studies found impairments [79,145,168,204] or trend toward impairments [205] in the WCST. Among studies reporting normal WCST performance, some had moderate effect sizes indicating possible impairments in TS patients [56,135,203]. These studies might have lacked the statistical power to detect statistically significant group differences.

Part B of the trail making test (TMT) can also yield significant evidence regarding cognitive flexibility. In some studies, children and/or adolescents with TS have been found to have poorer scores, in comparison with healthy controls [144,168,205]. In adults, impairments [34,57,98] or a trend toward an impaired performance [145] were also reported. Few studies also reported normal performance in TMT Part B, both in children [102] and adults [107,135,167]. Here again, some studies that did not report significant group difference had moderate effect sizes suggesting possibly impaired TMT performance in TS patients [102,206], and could therefore have lacked the sufficient power to detect this effect.

Cognitive tasks involving set-shifting also give an interesting insight on cognitive flexibility. In children with TS, opposing results have been found. In the intra-extra dimensional set-shifting subtest of the CANTAB, Rasmussen et al. [67] found an impairment, while Lin et al. [71] did not. Channon et al. [102] used a set-shifting task in which participants must respond to a card according to a rule in a first set of cards (“yes” to red, “no” to black), and then shift to a different rule in the next set (“yes” to cards of the same color as the previous one, “no” to cards of a different color). TS-only, TS + ADHD and TS + OCD children were not impaired in this task, but the authors indicated that it was simpler than the WCST. In a visual set-shifting task where patients had to make a movement according to the color of stimuli, Greimel et al. [62,75] found no specific impairment. Adults with TS made more errors in extra-dimensional shifting than healthy controls in an attentional set-shifting test including different stimuli dimensions (shape, color, number), [117]. In a number ordering task, where participants must switch between ordering them according to their magnitude and parity, adults with TS were impaired in total, mixing and switching accuracy [203].

Deficits in cognitive flexibility could be induced by the presence of comorbid disorders, and controlling this factor could result in no significant difference between TS patients and healthy controls [205]. For instance, OCS have been associated with worse WCST performance in TS patients [79,204,207]. However, normal WCST performance was also reported in patients with TS + OCD/OCS [56,137], which could be attributable to the small samples in each studies (*n* = 14). Comorbid ADHD could also be a confounding factor, but no consensus was reached across studies. For instance, some studies found no impact of ADHD on the cognitive flexibility of TS patients, measured either with the WCST [137,208] or the TMT [137]. Yet, better performance in the WCST [74] or the TMT [68,90] has also been reported in TS-only patients, when compared to TS + ADHD patients. Also, Brand et al. [148] found weaker but not statistically different TMT Part B performance in TS + ADHD patients, which could be attributable to low power. According to deGroot et al. [137], TS-only and TS + ADHD patients had normal WCST performance, but TS + ADHD + OCD patients were impaired. Taken together, these results suggest that comorbid OCD worsen TS patients’ cognitive flexibility, but the presence of ADHD could be problematic in some instances as well. Task complexity might also explain some of the discrepancies between studies. In the studies of Greimel et al. [62,75], the set-shifting tasks resembles Simon task, in which impairments are somewhat not frequent. In the study of Channon et al. [102], the set-shifting task is said to be easier than the WCST. Set-shifting deficits might, however, begin to appear as task complexity increase. 

#### 2.6.3. Planning Skills

Planning skills refer to the capacity to organize cognitive behavior in order to perform individual steps needed to attain a given objective [209]. In children with TS [73,89,166] or TS + ADHD [73], no significant deficits were found regarding Tower of London or Tower of Hanoï performance. However, another study reported that TS-only and TS + ADHD patients showed various impairments during the Tower of London test [87]. Yet, their TS-only and TS + ADHD groups were rather small (*n* = 13 and *n* = 8, respectively). While Mahone et al. [73] did not report a significant difference between TS + ADHD children and controls, the effect size between those two groups was rather large (Cohen’s d = 0.69). Their study might have lacked power to detect this group difference. In the Stockings of Cambridge test, which is also a tower test involving spatial planning, children with TS usually solve fewer problems than healthy controls [67,71]. However, in the study of Lin et al. [71], all children with TS had comorbid ADHD. In the six elements test, a planning test involving multitasking abilities, normal performance was reported in TS-only children, while TS + ADHD were impaired [102].

In adults, normal planning skills have been reported, measured either with the Tower of London [93,107,117] or the Six elements test [162]. These results suggest that planning skills are unimpaired in adults with TS. In children, the situation if less clear. Future studies are needed to make a definite ascertainment, but it is possible that ADHD impairs planning skills in children with TS. Therefore, planning skills might rely on good attentional capacities. Also, it seems plausible that the difference in planning abilities between TS patients and healthy controls are reduced as patients get older. And while neuropsychological studies do not report constant planning impairments, clinical questionnaires often report different organizational planning in TS and related disorders [210,211].

#### 2.6.4. Decision-Making

Hot executive functions are used to solve problems with an affective or motivational component [212], in areas such as decision-making [213]. In contrast, cold executive functions can be distinguished from hot executive functions by the absence of an affect linked to the cognitive process. Theoretically, hot executive functions mainly involve the activation of the ventromedial prefrontal cortex, while cold executive functions involve the activation of the dorsolateral prefrontal cortex [214]. Therefore, impairments in one do not necessarily imply impairments in the other [215]. Making such a distinction could be of particular importance in TS, since decision-making in these patients could be influenced by inherent impulsivity [130,131].

Few studies assessed decision-making in TS. Relative to healthy controls, children with TS were shown to prefer safer choices in a children-adapted version of the Iowa gambling task (IGT) [163]. In adults with TS, no IGT impairments were reported [88,179]. 

With other decision-making paradigms, results are not clear-cut. In the Cognitive bias test, adaptive decision-making did not differ between healthy controls and adults with TS [113]. Furthermore, counterfactual thinking, which implies the ability to generate alternative outcomes to past event or actions, was shown to be intact in adults with TS [216]. However, adults with TS had a minor deficit in selecting the most likely outcome in the Rogers decision-making task [117]. While more data is needed to make a definite statement regarding decision-making abilities, the available evidence suggests that they are not significantly impaired in TS.

#### 2.6.5. Extent of Executive Functioning Impairment in TS

In TS, impairments in executive functions appear to be quite specific, rather than widespread. Studies report more frequent deficits regarding inhibitory control and cognitive flexibility, while impairments in planning and decision-making seem to be somewhat less frequent. Yet, cognitive flexibility involves inhibitory control, and it remains to be determined how the inhibitory deficits of TS patients impair their cognitive flexibility. Available data also showed that comorbid ADHD or OCD can be detrimental to TS patients’ executive functions, especially regarding inhibition and cognitive flexibility.

### 2.7. Social Cognition

Social cognition refers to the cognitive processes used by individuals in their interactions with others [217]. It allows for adequate social adjustment and functioning [218]. Social cognition involves basic social functions (e.g., face recognition, emotion perception) as well as higher-order processes (e.g., social judgment, empathy, moral reasoning) [219] that are mediated by specialized areas such as the fusiform gyrus [220], as well as complex neural networks involving the basal ganglia and frontal lobes [221], for example. In comparison with other neuropsychological domains, social cognition has not been studied broadly in TS, but interesting findings have started to emerge in the last few years. Since TS and autism spectrum disorders (ASD) sometimes co-occur and share clinical features [222], a recent study aimed to examine ASD symptoms in TS patients [223]. This study reported that ASD symptoms in TS (measured as the ability to engage in emotionally appropriate reciprocal social interactions) were higher than in the general populations, but were comparable to the prevalence among other psychiatric disorders. Furthermore, ASD symptoms were more present in children than adults with TS, and were strongly related to comorbid OCD. Such findings of elevated ASD symptoms among TS patients are consistent with the hypothesis of slight impairments in social cognition.

In children and adolescents with TS, social responsiveness has been shown to be impaired, which suggests autistic-like symptoms in those patients [205], and is consistent with the study of Darrow et al. [223]. However, those social impairments could be attributable to specific tics, comorbid disorders or cognitive impairments rather than TS itself. In children with TS + ADHD, mild deficits in emotion recognition were reported, especially for items involving anger [91].

A recent review showed that adults with TS had mild deficits in generating and rating solutions aimed at resolving tricky social situations and that they may exhibit unconventional interpretations of material with humorous content [224]. Small differences in the interpretation of facial expressions were also found, and stimuli involving conflicting emotions seemed to be more likely to elicit unconventional reactions. Yet, emotional recognition based on partial cue didn’t differ in another study between adult with and without TS [225]. TS patients also had different reasoning in the Faux pas task [96]. However, there was no problem in understanding false beliefs. Another study also found that Faux-pas detection seems impaired in TS-only adults, yet they showed no deficit in explaining the Faux-pas they correctly identified and showed intact emotional self-disclosure [226].

Similarly to children with TS + ADHD, adults with TS-only also struggled with anger items in emotional recognition tasks [91]. Drury et al. [227] found that TS-only adults were more associated with suppression emotion regulation strategies (an inhibition strategy) than a group of healthy controls. Yet, both groups reported the same level of emotional expression and did not differ on the use of reappraisal strategies. When presented with ambiguous visual stimuli (moving triangles), TS adults patients tended to hyper-mentalize and attribute more intentionality to stimuli’s movements, but they showed conventional answer on goal-directed animations and on theory of mind animations [98]. Eddy et al. [228] used the interpersonal reactivity index to assess empathy in adults with TS. Their results showed that TS is associated with a reduced tendency to adopt other people’s perspective. Those differences in social cognition were not related to comorbid disorders such as ADHD or OCD, suggesting that impaired social cognition could be inherent to TS. Yet, TS-alone adults did not differ from healthy control on tasks involving explanation of others actions and on cognitive and emotional empathy task [162].

Contradictory findings do point toward a slight impairment within this population. Little information is available on younger TS patients and most reported adult studies sampled less than 25 patients with TS. More studies are needed to fully understand the social cognition impairments of TS patients. 

### 2.8. Learning Difficulties and Disabilities

Neuropsychological evaluation often includes assessing the presence of learning disabilities and their accompanying neurocognitive deficits. Although poor academic achievement has been reported for a long time in TS [229,230], the relationship between TS and learning disabilities remains unclear. This may be in part because learning disabilities have been used as a synonym to learning difficulties and learning problems. From this point of view, many factors other than cognition have been advanced as barriers to learning in TS, such as comorbidities and environmental elements. Here, we use the term learning difficulties to refer to poor academic achievement, and learning disabilities as referring to specific learning disorders such as dyslexia and dyscalculia.

Learning difficulties in TS have been studied as the result of the negative impact of tics on daily functioning. In special education and learning disability classes, the prevalence of tic disorders is approximately 10 times higher than in the general population [231]. In many cases, learning difficulties are associated with delays in learning skills, poor attention, hyperactivity, or other behavioral and/or psychological disorders [229,230,232]. Tics can be distracting and intense to the point where they impede the ability to keep eyes on the paper to read [233]. A recent study revealed that high school students with TS face difficulties in doing school work and managing emotions in school [234]. This study also reported frequent concentration problems, suggesting that ADHD could partly explain school difficulties in TS. Slow learning and attention problems were also reported in high school students with TS as frequent difficulties encountered in school [7]. Learning problems in reading, comprehension, spelling, and math have been identified in 30 to 40% of cases of individuals with tic disorders [235,236]. Furthermore, arithmetic deficiencies have been found in children with TS [128], and were correlated with attentional deficits in a CPT [66]. However, children without CPT impairments had no arithmetic problems. In a recent study on educational problems in TS, it was reported that more than half of TS patients needed some sort of extra-educational support, and that the presence of comorbid disorders was associated with more educational problems [237].

Although it is generally agreed that a link exists between TS and school difficulties, Cubo et al. [238] did not find any association between tic disorders and poor academic performance in a population-based study of school-aged children in Spain. Another study also reported that most parents of children with TS claimed that their child had an above average academic performance [239]. Therefore, discrepancies exist, but it seems clear that a proportion of children with TS experience academic-related problems. The presence of comorbid disorders, the use of psychiatric medication and the role played by school personnel could influence learning difficulties in TS. Also, such discrepancies could rely on the fact that if effective behavioral and integration strategies are put in place in schools, then the academic potential of TS children is not altered and learning strategies are working well.

In regards to learning disabilities, their prevalence in TS have been reported to be specific to math and written language [37]. The prevalence of learning disabilities in TS usually sits at around 20% [11,240], but they are more frequent in children and in patients with comorbid ADHD [11]. A cohort study showed that 11% of non-comorbid TS and 31% of TS + ADHD children had various school difficulties, which mainly regarded reading and handwriting [241]. Recently, Neri et al. [232] have observed that 15% of their sample of 3 to 12 years old children with TS presented with a comorbid learning disability such as dyscalculia, reading, or spelling disorder, which was associated with worse functioning then in TS alone. Burd et al. [9] identified five factors that increase the risk of having learning disabilities in TS patients: being male, receiving a TS diagnosis at a young age, having fewer family members with tics, having perinatal problems, and having more comorbidities. 

Whether the academic functioning results from TS itself or from a specific learning disability, much remains to be learned about the relation between TS and learning difficulties/disabilities. Research so far highlights the importance of evaluating academic performance, screening for learning disabilities and considering the impact of tics on learning skills. 

## 3. Conclusions

This review allowed a synthesis of the important findings regarding the neuropsychological functions of TS patients. It also permitted us to understand how various factors could impact these functions and therefore confound the results of prior studies. Table 1 gives an overview of the proportion of studies within a neuropsychological dimension that controlled for such confounding factors, and summarizes the main conclusions of the review.

### 3.1. Recent Advances

In the last decade, neuropsychological research has been flourishing in the field of TS. In their review, Eddy et al. [36] noted that multiple confounding factors could affect TS patients’ performance in neuropsychological tests, which could account for discrepancies between studies. To circumvent this problem, recent research has tended to use improved and more sound methodology, with a better control of comorbidity and medication. 

Prior evidence suggested that attentional capacities could be influenced by comorbid disorders [36]. Research from the last decade seems to confirm this assumption, as attentional deficits have been found in TS + ADHD [62,69] or TS + OCD [77] patients. Yet, it remains to be confirmed whether TS-only patients show constant attentional deficits or not.

As reported by Eddy et al. [36], only slight memory impairments seem to exist in TS, and most impairment could be attributable to comorbid disorders. Furthermore, a recent study revealed that increased tic severity was associated with impaired memory [100].

Motor skills have not been studied extensively in recent years. A recent review reported that studies on this matter failed to reach a conclusive outcome, citing reasons such as comorbidity, patients’ age and control measures to explain discrepancies [138]. Yet, more recent research with the one of the largest sample to date among neuropsychological studies of TS patients found a fine and gross motor dexterity performance that was one SD below the mean of normative data [134]. This suggests that prior studies might have lacked the power to detect such impairments in TS patients.

In regards to executive functioning, recent research has proven the existence of inhibitory deficits in TS patients [130,203]. While previous reviews assured that TS patients performed poorly in sentence completion tasks [36,161], we can now confirm that CPT and Stroop performance are also impaired in TS patients. Such impaired inhibitory control could therefore alter TS patients’ performance in other tasks, and could possibly account for deficits in cognitive flexibility that were sparingly reported in the last decade [98,145,168,203,205]. A lack of cognitive flexibility diminishes the adaptation to new situations and it affects social and emotional behaviors, which can lead to rigid behavior, anxiety, and depressive symptoms [205]. Few recent studies focused on planning and decision-making, and these functions seem relatively intact in TS patients. Yet, comorbidity could possibly cause some impairments.

While Eddy et al. [36] suggested that TS patients might exhibit verbal fluency deficits, most recent data appear to contradict this, as normal letter [34,36,86,144,145] and categorical [34,145,147] fluency were often reported in contemporary data. Yet, impaired letter fluency has been reported in a recent study with a large sample of TS-only patients, so more research on fluency is still needed to clearly understand this function. Others aspects of language production and comprehension do not appear to be extensively affected in TS patients.

Our review sheds light on important domains that were not part of the review of Eddy et al. [36]: social cognition and academic performance. Research in these domains has been particularly flourishing in recent years. Contemporary data on social cognition suggest slight impairments in this domain, mostly in emotional regulation [91,224,227] and social interactions [205,223] in TS patients. In regards to school difficulties, recent research suggests that learning difficulties and disabilities could occur more frequently in children with TS, but more research in this domain is needed to make strong assumption.

### 3.2. Limitations

This study has some limitations. We opted to conduct a narrative review, which is not as rigorous as a systematic review. Yet, we believe that the whole field of the neuropsychology of TS might be too large to be reviewed systematically, and that systematic reviews and/or meta-analyses focusing on a single function (e.g., attention, cognitive flexibility, etc.) would be more relevant. This review is also limited by the amount of literature on given domains. For example, there are only few available studies on the social cognition of children with TS, which makes it hard to draw clear conclusions on this matter. We encourage researchers to conduct studies in such domains where few data are available.

### 3.3. Clinical Implications

Impairments reported in this review concern TS patients’ performance during neuropsychological tasks. However, these impairments are reflected in the daily functioning of TS patients. For example, impaired inhibitory control might lead to more impulsive manifestations [130], such as conduct disorders or explosive outbursts [242,243]. Impairments in cognitive flexibility could lead patients to make their decision in a more rigid fashion, which could in turn lead to insecurity and anxiety [163].

Also, various impairments across neuropsychological functions may impact TS children’s activities at school. For instance, problems in inhibition or flexibility could be detrimental to relations with peers, deficits in planning capacities could lead to difficulties during exams and attentional deficits could impair proper encoding of important knowledge. Such deficits should be considered when dealing with TS children in school settings. The use of proper behavioral strategies could decrease the impact of neuropsychological impairments and allow good academic performance.

All in all, patients with TS do not show major neuropsychological impairments. The profile does not differ much between children and adults, as slight deficits have been reported throughout the lifespan in nonverbal memory, motor skills, inhibition, flexibility and social cognition. Yet, since TS severity is known to decrease in adulthood, the selection of adults with significant symptoms by the studies reported here could have induced a bias regarding the neuropsychological profile of those adults. Longitudinal studies could shed light on the evolution of neuropsychological functions from childhood to adulthood.

### 3.4. Future Directions

While the latest studies allowed a better understanding of neuropsychological functioning of TS patients, much remains to be explained. Future studies should follow the trend set in recent years and continue to publish research with strong methodology (i.e., control of comorbidity and medication, large samples, etc.). We also encourage researchers to conduct meta-analyses where much data is available regarding a specific neuropsychological test or function (e.g., WCST, TMT, Purdue Pegboard).

Yet, if studies with better methodology continue to produce discrepant results, a paradigm change might be needed. We should maybe consider cognitive and behavioral profiles in a transnosological approach to allow the identification of subgroups that may be more representative, given the frequent discrepancies in results. We hypothesize that the complexity and the heterogeneity of TS could be better understood by identifying cognitive profiles beyond DSM categorical diagnoses. This idea is in line with the research domain criteria (RDoC), an innovative approach for classifying mental disorders implemented by the NIMH [244,245]. Alternatively, Dell’Osso et al. [14] proposed the phenotype of obsessive–compulsive tic disorder (OCTD), which emerges from the new “tic-related” specifier for OCD introduced in the DSM-5. Given the fact that TS and OCD are deeply interconnected and could represent a specific subtype of illness, the authors argue that identifying commonalities between both disorders could provide helpful insight for patients and clinicians.

A goal of this review was to untangle the cognitive impairments attributable to TS itself from those caused by comorbid disorders. Yet, comorbidity is often present in TS, and the neuropsychological profile of a TS-only patient might not be representative of a typical TS patient with various comorbidities. Therefore, one could question the necessity of making this distinction. However, it seems useful to know how neuropsychological deficits are related to specific comorbid disorders, in order to provide individualized treatment for TS patients.

## Figures and Tables

**Table 1 brainsci-07-00106-t001:** Control of confounding factors in experimental studies and main findings in each neuropsychological dimension.

Neuropsychological Dimensions	Studies with Children	Studies that Excluded or Controlled for Comorbid ADHD	Studies that Excluded or Controlled for Comorbid OCD	Studies that Excluded or Controlled for All Comorbidities	Studies that Excluded or Controlled for Medication	Summary of Main Findings
Intellectual abilities	100%	100%	60%	40%	20%	Only slight impairments in intellectual abilities seem to exist in children with TS. Comorbid disorders could affect the extent of those impairments, but this remains to be confirmed. Studies on this matter should also be conducted in adults with TS.
Attention	74.1% *	63.0%	40.7%	11.1%	44.4%	Only slight impairments were found regarding TS patients’ attentional abilities. Tic severity, comorbid ADHD, and task complexity could impair TS patient’s performance.
Memory	58.6% *	69.0%	62.1%	31.0%	48.3%	Some slight deficits could exist regarding nonverbal memory, but globally, TS patients do not have major memory impairments.
Motor skills	62.5% *	53.1%	43.8%	6.3%	57.6%	It is hard to draw conclusions regarding TS patients’ motor skills. Comorbidity could potentiate some small inherent deficits. Visual motor integration and visuoconstructive skills do not appear to be widely impaired either.
Language	65.4% *	57.7%	42.3%	11.5%	46.2%	Verbal fluency seems intact in TS patients, but confounding results have been reported. Comorbidity and tic severity could have an impact. Small impairments in language comprehension could be linked to deficits in social cognition.
Executive functions	62.2% *	71.6%	56.8%	21.6%	48.6%	Executive functions deficits appear to be limited to inhibition and cognitive flexibility. Impairments in cognitive flexibility could be attributable to poor inhibition. Planning and decision-making abilities seem intact.
Social cognition	22.2% *	66.7%	66.7%	44.4%	44.4%	Few data are available regarding social cognition in TS, but slight impairments seem to exist, which would be consistent with some autistic traits found in TS. More studies on social cognition should be conducted in children with TS.

Note: * Some studies included both children and adults.

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
