# Peer review of "A Review of the Neuropsychological Dimensions of Tourette Syndrome"

_brainsci, 2017, doi:10.3390/brainsci7080106_

Round 1

Reviewer 1 Report

Dear authors, 

your manuscript is original by contents and it is very complete, unless TS picture strictly depends on its comorbidities, age of the patient and other features such as IQ. 

Anyway, some minor edits are needed.

1) What about the precise method of the review? how did you select papers? which is the source (e.g. Pubmed and others?) how did you choose criteria of selection of papers? which is the historical period of the selection of papers?

2) Some specific revisions are: 

- line 32 and on "confounding factor": maybe it is better to explain in which sense factors can be confounding. This should be done at the beginning of the paper.

- line 42 typing mistake: "waxe and wane"

- line 43: up to 20% if considering Tic Disorders, in general see  Bloch and Leckman, 2009

- line 86: up to 90% considering OCS (Gomes De Alvarenga et al. 2012; Ferrao et al. 2013)

- line 91: even CBT and DBS can influence neuropsychological studies

- line 230: CBT and DBS should also be considered as treatments influencing neuropsychological performances

- line 253: delete one "the"

- line 273 "moderate tic symptoms.." can be useful to add YGTSS scores, in these cases?

- line 433: can sarcasm deficit be linked with OCD traits?

- line 479: "RT" stands for response time? Please, specify it.

- line 771: you can have add a note about Dell'Osso et al's "Parsing the phenotype of obsessive-compulsive tic disorder (OCTD): a multidisciplinary consensus", published in 2017.

- line 775: "by" instead of "be".

3) I think you should consider anxiety and depression TS coexisting factors, as well, as influencing cognitive factors, in children and in adults.

4) as you wrote about school difficulties you may add a paragraph about work-related difficulties in adults, including relationships with colleagues. 

Author Response

Reviewer 1

Reviewer’s comment: “Dear authors, 

your manuscript is original by contents and it is very complete, unless TS picture strictly depends on its comorbidities, age of the patient and other features such as IQ. 

Anyway, some minor edits are needed.”

Response: We wish to thank the reviewer for his valuable comments and suggestions, which helped to improve our manuscript. Each comment was addressed and corrections were made accordingly in the text.

Reviewer’s comment: “1) What about the precise method of the review? how did you select papers? which is the source (e.g. Pubmed and others?) how did you choose criteria of selection of papers? which is the historical period of the selection of papers?

Response: Since our manuscript is a narrative review, rather than a systematic review, there was no precise methodology used to select the studies that are discussed in our manuscript. We believe that the whole field of neuropsychology of TS might be too large to be reviewed systematically, and that systematic reviews focusing on a single function (e.g.: attention, cognitive flexibility, etc.) would be more relevant. However, we added this as a limitation (lines 772-775).

Reviewer’s comments:

2) Some specific revisions are:

- line 32 and on "confounding factor": maybe it is better to explain in which sense factors can be confounding. This should be done at the beginning of the paper.

Response: We added a sentence to explain in which sense these factors can be confounding: “These confounding factors can be understood as the important heterogeneity in patients’ symptoms and associated features, which are not always controlled in some studies.” (lines 90-92).

“- line 42 typing mistake: "waxe and wane"

            Response: Done.

- line 43: up to 20% if considering Tic Disorders, in general see  Bloch and Leckman, 2009

Response: We added a reference to express that the prevalence of other tic disorders is higher (lines 43-44).

- line 86: up to 90% considering OCS (Gomes De Alvarenga et al. 2012; Ferrao et al. 2013)

Response: We added a mention to expose the fact that OCS can occur more frequently than clinical OCD in TS patients (lines 55-58).

- line 91: even CBT and DBS can influence neuropsychological studies

Response: We added a sentence and a few references to express this: “Other treatments, such as cognitive-behavioral therapy (CBT) or deep-brain stimulation (DBS), can affect neuropsychological functions as well…” (lines 100-101).

- line 230: CBT and DBS should also be considered as treatments influencing neuropsychological performances

            Response: This was answered in regard to the previous comment (lines 100-101).

- line 253: delete one "the"

            Response: Done.

- line 273 "moderate tic symptoms.." can be useful to add YGTSS scores, in these cases?

            Response: We added the mean YGTSS score for each group (lines 282-284).

“- line 433: can sarcasm deficit be linked with OCD traits?”

Response: In the cited study, it is mentioned that there is no relation between sarcasm comprehension and OCD symptoms. We added a sentence to express this: Authors mentioned these impairments are unlikely to be influenced by TS symptomatology, comorbid disorders, or medication intake (lines 446-447).

“- line 479: "RT" stands for response time? Please, specify it.”

            Response: Specified (line 490).

“- line 771: you can have add a note about Dell'Osso et al's "Parsing the phenotype of obsessive-compulsive tic disorder (OCTD): a multidisciplinary consensus", published in 2017.”

Response: We thank the reviewer for this suggestion. Indeed, this article is relevant to our manuscript and we added a reference to it (lines 813-817).

“- line 775: "by" instead of "be".”

            Response: Done.

Reviewer’s comment: “3) I think you should consider anxiety and depression TS coexisting factors, as well, as influencing cognitive factors, in children and in adults.”

Response: Response: Although this is not directly related to our main topic, precisions have been added to the introduction and conclusion regarding these other co-occurring conditions (lines: 50-51; 96; 753-755; 784-786).

Reviewer’s comment: “4) as you wrote about school difficulties you may add a paragraph about work-related difficulties in adults, including relationships with colleagues.”

Response: This is indeed an interesting topic, although we have decided not to address it as it goes beyond the scope of this paper, contrary to school difficulties, which are tied up to learning disabilities, which in turn are difficult to avoid when addressing neurocognition. To clarify this point, we renamed the section 2.8 “Learning difficulties and disabilities” instead of “School difficulties”.

Reviewer 2 Report

This review of neuropsychological performance in patients with Tourette Syndrome (TS) is an effort to summarize the conflicting reports in the literature.  The manuscript meets a need of the field to rectify disparate findings, considering not all studies accounted for the confounds of age, medication status and comorbidity status.  If clarified, a better picture of neuropsychological functioning in pure TS will be known, allowing more effective, timely interventions.

Comments are below.

Major

In the introduction, the authors rightfully point out the confounders of comorbidities, medication status and age.  Yet, it is confusing in the rest of the review how many studies properly accounted for them.  Please consider a table listing what percent of papers in each neuropsychological domain accounted for each of these confounders.

The introduction states that because the last review of neuropsychological performance in TS was published in 2009, the field needs this review that will confirm whether better-designed studies in the last 10 years can clarify discrepant findings.  With the additional mention of the first aim (to update the state of knowledge in the field), readers can be led to believe this review will only include research published between 2009 and 2017.  This is not the case.  Please clarify this in the introduction.

Many tests of neuropsychological function are reviewed, but eye movement measures (e.g., saccade, antisaccade, working memory) are not included.  Please include a review of these findings, which can fall under the current headings of attention, working memory, inhibition, etc.

The review is text-heavy, with no visual summary of the information.  Please include a table that encapsulates the conclusions of the text.  This could be combined with the tabular information suggested in point 1 above.

Minor

Please check for several noun/verb disagreements.  For example, the abstract reads, “neuropsychological functions evolves.”

Please check for missing words (e.g., “wax and wane”) and correct grammar.

Once an acronym is defined, continue to use it subsequently throughout the manuscript.

Please be accurate in statements.  For example, on page 15, line 712, study authors cannot increase tic severity intentionally.  Rather, the study likely compared increased tic severity to moderate tic severity and controls.

Author Response

Reviewer 2

Reviewer’s comment: “This review of neuropsychological performance in patients with Tourette Syndrome (TS) is an effort to summarize the conflicting reports in the literature. The manuscript meets a need of the field to rectify disparate findings, considering not all studies accounted for the confounds of age, medication status and comorbidity status. If clarified, a better picture of neuropsychological functioning in pure TS will be known, allowing more effective, timely interventions.

Comments are below.”

Response: We wish to express our gratitude to the reviewer for his comments that allowed to significantly improve our manuscript. Each comment was addressed, and the manuscript was modified according to the reviewer’s comments and suggestions.

Major

Reviewer’s comment: “In the introduction, the authors rightfully point out the confounders of comorbidities, medication status and age.  Yet, it is confusing in the rest of the review how many studies properly accounted for them.  Please consider a table listing what percent of papers in each neuropsychological domain accounted for each of these confounders.”

Response: We thank the reviewer for this suggestion. It is true that it might help the reader to know the percent of papers that accounted for confounding factors. We added a table in the conclusion that lists the proportion of studies that accounted for these confounding factors (lines 730-734 and Table 1).

Reviewer’s comment: “The introduction states that because the last review of neuropsychological performance in TS was published in 2009, the field needs this review that will confirm whether better-designed studies in the last 10 years can clarify discrepant findings.  With the additional mention of the first aim (to update the state of knowledge in the field), readers can be led to believe this review will only include research published between 2009 and 2017.  This is not the case.  Please clarify this in the introduction.”

Response: Indeed, some readers could have been led to believe that our review only focused on post-2009 papers. To resolve this problem, we added a sentence before the mentions of the review’s aims: By merging recent advances with prior knowledge, this review will yield a comprehensive picture of the neuropsychological functions of TS patients.” (lines 119-121).

Reviewer’s comment: “Many tests of neuropsychological function are reviewed, but eye movement measures (e.g., saccade, antisaccade, working memory) are not included.  Please include a review of these findings, which can fall under the current headings of attention, working memory, inhibition, etc.”

Response: We added a section regarding oculomotor studies in the section on inhibition: “Oculomotor studies may also yield pertinent information about response inhibition. On the one hand, some studies reported increased latencies [100,191-193] or higher error rate [191,193,194] for TS patients during antisaccade tasks. Surprisingly, tic severity is unlikely to affect oculomotor response inhibition, since Jeter et al. [100] reported a trend toward longer latencies for TS children with low severity when compared to children with moderate severity. On the other hand, enhanced cognitive control in antisaccade tasks has also been reported. In the study of Tajik-Parvinchi and Sandor [195], the TS+ADHD+OCD patients had shorter antisaccade latencies than healthy controls, TS-only and TS+ADHD patients, and made less errors than healthy controls and TS+ADHD patients. Some studies also reported that children with TS could switch between prosaccade and antisaccade tasks faster than healthy controls [196,197]. Another study found similar switching latencies for TS children and healthy controls, but better switching accuracy for TS children [198]. Therefore, more research is needed to better understand how TS patients perform during oculomotor paradigms.” (lines 510-521).

Reviewer’s comment: “The review is text-heavy, with no visual summary of the information.  Please include a table that encapsulates the conclusions of the text.  This could be combined with the tabular information suggested in point 1 above.”

Response: We thank the reviewer for this suggestion. A summary of the main findings regarding each neuropsychological dimension was added in the table that reports the proportion of studies accounting for confounding factors (see Table 1)

Minor

Reviewer’s comment: “Please check for several noun/verb disagreements.  For example, the abstract reads, “neuropsychological functions evolves.”

Response: We corrected this mistake and revised the manuscript to check for other noun/verb disagreements.

Reviewer’s comment: “Please check for missing words (e.g., “wax and wane”) and correct grammar.”

            Response: This was corrected.

Reviewer’s comment: “Once an acronym is defined, continue to use it subsequently throughout the manuscript.”

Response: We revised the manuscript to assure that acronyms are correctly defined at first occurrence, and that they are used consistently throughout the manuscript.

Reviewer’s comment: “Please be accurate in statements.  For example, on page 15, line 712, study authors cannot increase tic severity intentionally.  Rather, the study likely compared increased tic severity to moderate tic severity and controls.”

Response: We wish to thank the reviewer for pointing this out. It appears that a part of this sentence was removed accidentally. This sentence should have read as follows: “Furthermore, a recent study revealed that increased tic severity was associated with impaired memory (lines 740-741)”

Reviewer 3 Report

In the current study, the authors conducted a review on the neuropsychological functions of TS. It reviews the factors that play a role in modulating these dimensions. The rationale is well organized and the findings are well put into perspective with previous work. This paper represents very meticulous work. However, this study has potential to contribute to the present knowledge of Tourette research literature, I have a few concerns for the manuscript:

1)    The search criteria for the studies are not clearly noted. A supplement could be provided giving all the search terms and search engines used. Inclusion and exclusion of studies reviewed should also be noted.

2)    As described by the authors, the review aims at resenting factors impairments of neuro-cognitive functions. However, they have mainly focussed on age and co-morbid OCD and ADHD. There were no discussions about the role of other commonly associated co-occurring conditions like anxiety, depression, depressive symptomatology, oppositional, conduct disorders, etc.

3)    The quality of writing is good, but there are a few typing errors in the sentences which need to be rectified. For example, Line 42: “Wax and [2]” ; Line [530] “In a visual set shifting task were patients had to…”

4)    In Tourette syndrome research terms Obsessive Compulsive Disorder (OCD) and Obsessive Compulsive Symptoms (OCS) have been used interchangeably. Are there any differences between these terms and their outcome on cognitive profiling of patients? If so, these may require extended description for audiences that do not have a background in clinical research. It would be ideal to differentiate them in one or two sentences.

5)    In addition to developmental age and comorbid conditions, does gender bias play any role in modulating the impairments in neuro-cognitive functions? It has been reported that TS is more common in males than in females. Also, males are more likely to have higher rates of comorbidities than women. I speculate that this could also be a confounding factor in modulating the cognitive functioning.

6)    Please check the sub-heading :2.6.5 Executive functions under 2.6 Executive functions. Why do we need it?

7)    In the end, I would like to ask authors to add a paragraph listing limitations of the review. Explaining the nature of the limitations, justify the choices that they have made during the review process. Finally, suggesting how such limitations could be overcome in future.

Author Response

Reviewer 3

Reviewer’s comment: “In the current study, the authors conducted a review on the neuropsychological functions of TS. It reviews the factors that play a role in modulating these dimensions. The rationale is well organized and the findings are well put into perspective with previous work. This paper represents very meticulous work. However, this study has potential to contribute to the present knowledge of Tourette research literature, I have a few concerns for the manuscript:”

Response: We wish to thank the reviewer for his comments, which helped us to significantly improve our manuscript. All concerns were addressed and the manuscript was adapted according to the reviewer’s comments.

Reviewer’s comment: “1)    The search criteria for the studies are not clearly noted. A supplement could be provided giving all the search terms and search engines used. Inclusion and exclusion of studies reviewed should also be noted.”

Response: Since this review is a narrative review, rather than a systematic review, there was no systematic methodology used to retrieve relevant papers. We believe that the whole field of neuropsychology of TS might be too large to be reviewed systematically, and that systematic reviews focusing on a single function (e.g.: attention, cognitive flexibility, etc.) would be more relevant. However, we added this as a limitation.

Reviewer’s comment: “2)    As described by the authors, the review aims at resenting factors impairments of neuro-cognitive functions. However, they have mainly focussed on age and co-morbid OCD and ADHD. There were no discussions about the role of other commonly associated co-occurring conditions like anxiety, depression, depressive symptomatology, oppositional, conduct disorders, etc.”

Response: Response: Although this is not directly related to our main topic, precisions have been added to the introduction and conclusion regarding these other co-occurring conditions (lines: 50-51; 96; 753-755; 784-786).

Reviewer’s comment: “3)    The quality of writing is good, but there are a few typing errors in the sentences which need to be rectified. For example, Line 42: “Wax and [2]” ; Line [530] “In a visual set shifting task were patients had to…”

            Response: We thank the reviewer for pointing this out. Corrections were made accordingly.

Reviewer’s comment: “4)    In Tourette syndrome research terms Obsessive Compulsive Disorder (OCD) and Obsessive Compulsive Symptoms (OCS) have been used interchangeably. Are there any differences between these terms and their outcome on cognitive profiling of patients? If so, these may require extended description for audiences that do not have a background in clinical research. It would be ideal to differentiate them in one or two sentences.”

Response: We added a few sentences in the introduction to explain both terms and to express how TS patients are affected (lines 55-58).

Reviewer’s comment: “5)   In addition to developmental age and comorbid conditions, does gender bias play any role in modulating the impairments in neuro-cognitive functions? It has been reported that TS is more common in males than in females. Also, males are more likely to have higher rates of comorbidities than women. I speculate that this could also be a confounding factor in modulating the cognitive functioning.”

Response: We would like to thank the reviewer for pointing out this important issue, which we have added in the list of potential confounding factors (lines 103-105).

Reviewer’s comment: “6)    Please check the sub-heading :2.6.5 Executive functions under 2.6 Executive functions. Why do we need it?”

Response: Since the section on executive functions is rather large, we added this subsection to integrate the findings in a few sentences. However, we agree that it might be misleading to have two sections with similar headings. We therefore changed the heading of subsection 2.6.5 to “Extent of executive functioning impairment in TS” (line 619).

Reviewer’s comment: “7)    In the end, I would like to ask authors to add a paragraph listing limitations of the review. Explaining the nature of the limitations, justify the choices that they have made during the review process. Finally, suggesting how such limitations could be overcome in future.”

Response: We added a new section listing the limitations of our review (section 3.2, lines 771-779).

Round 2

Reviewer 2 Report

The authors have addressed each of the reviewers' concerns.  I am especially pleased with the addition of Table 1.